# From Melanocytes to Melanoma Cells: Characterization of the Malignant Transformation by Four Distinctly Different Melanin Fluorescence Spectra (Review) [note 1]

**DOI:** 10.3390/ijms22105265

**Published:** 2021-05-17

**Authors:** Dieter Leupold, Lutz Pfeifer, Maja Hofmann, Andrea Forschner, Gerd Wessler, Holger Haenssle

**Affiliations:** 1LTB Lasertechnik Berlin, Am Studio 2c, 12483 Berlin, Germany; lutz.pfeifer@ltb-berlin.de; 2Klinik für Dermatologie, Venerologie und Allergologie, Charité-Universitätsmedizin, Charitéplatz 1, 10115 Berlin, Germany; maja.hofmann@charite.de; 3Hautklinik, Universitäts-Klinikum Tübingen, Liebermeisterstraße 25, 72076 Tübingen, Germany; andrea.forschner@med.uni-tuebingen.de; 4Hautarztpraxis Berlin, Linderhofstrasse 20, 12623 Berlin, Germany; gerd.wessler@web.de; 5Hautklinik, Universitätsklinikum Heidelberg, Im Neuenheimer Feld 440, 69120 Heidelberg, Germany; Holger.Haenssle@med.uni-heidelberg.de

**Keywords:** melanin fluorescence, melanoma subtypes, dysplastic nevi, dermatofluoroscopy

## Abstract

The melanin fluorescence emitted by pigment cells of the human skin has been a central research topic for decades, because melanin, on the one hand, protects against (solar) radiation in the near-UV range, whereas on the other hand, melanocytes are the starting point for the malignant transformation into melanoma. Until recently, however, melanin fluorescence was not accessible in the context of conventional spectroscopy, because it is ultraweak and is overshadowed by the more intense so-called autofluorescence of endogenous fluorophores. The advent of a new method of laser spectroscopy has made this melanin fluorescence measurable in vivo. A stepwise two-photon absorption with 800 nm photons is used, which more selectively excites melanin (dermatofluoroscopy). Our review summarizes the experimental results on melanin fluorescence of the four types of cutaneous pigment cells from healthy and malignant tissues. Outstanding is the finding that different types of melanocytes (i.e., melanocytes of common nevi, versus dysplastic nevi or versus melanoma cells) show characteristically different fluorescence spectra. The possibilities of using this melanin fluorescence for melanoma diagnosis are shown. Moreover, the uniform fluorescence spectra emitted by different melanoma subtypes are essential. Conclusions are drawn about the molecular processes in the melanosomes that determine fluorescence. Finally, experimental suggestions for further investigations are given.

## 1. Introduction

As a skin pigment, melanin may be viewed as a double-edged sword: on the one side, melanin acts as a parasol protecting epidermal stem cells from UV-induced DNA damage, and on the other side, the melanocytes that produce melanin carry the risk of transformation into melanoma, one of the most aggressive and deadliest forms of skin cancer. Melanin exerts its protective effect on the normal pigment cells, which are termed “melanocytes.” On the way from aggregated pigment cells (nevomelanocytes) of benign/common nevi or nevomelanocytes of dysplastic nevi towards melanoma cells, the effect of melanin shifts away from its original benefits to malignancy [1].

The objective characterization of these four subtypes of melanocytes with spectroscopic methods has been the goal of extensive research for decades, because a clear-cut differentiation either in vivo (directly on patient skin) or ex vivo (on tissue specimen after surgical excision) could be of great benefit for the diagnosis of melanoma.

In the past, much attention was paid to the fluorescence of melanin with the most crucial question to be answered: Is it possible to specifically measure melanin fluorescence in the skin and is it possible to differentiate the four subtypes of melanocytes in this way?

With regard to melanoma cells, the question was expanded, because “melanoma” presents itself in a variety of subtypes, which differ according to growth patterns, anatomical localization, and patterns of genetic aberrations [2,3]. Melanoma subtypes also differ with respect to the prognosis and outcome for the patient, mostly depending on their different potential to form metastases. The most important subtypes (with decreasing incidence rates) are the superficial spreading melanoma (SSM), lentigo maligna melanoma (LMM), acral lentiginous melanoma (ALM), and nodular melanoma (NM). In the context of classic, conventional spectroscopy, the fluorescence of melanin in skin tissue is not a suitable source of information because it is extremely weak and it is masked by the much more intense autofluorescence of endogenous skin fluorophores such as NADH and flavins [4,5,6,7].

A breakthrough in terms of the measurability of melanin fluorescence in tissue came with a special technique of non-linear spectroscopy, the stepwise two-photon excitation of fluorescence [7,8,9,10,11,12]. Its specialized version for measurements in tissue is called dermatofluoroscopy [13,14,15].

With dermatofluoroscopy, the ultraweak melanin fluorescence can be measured without interference from background autofluorescence. It turns out to be a meaningful source of information on the issues addressed above:Differentiation/characterization of the four types of pigmented skin cellsDetectability of all melanoma subtypes

On this basis, a new and objective melanoma diagnosis is possible both in vivo directly on the patient skin [16,17,18,19,20] and ex vivo on the histological specimen [11,21,22]. Until today, after measurements in hundreds of patients, there are no observations of any adverse effects by the photon flux densities of the stimulating laser radiation used for dermatofluoroscopy [23].

The results of the most important studies investigating dermatofluoroscopy are summarized below. In addition, we demonstrate that measuring the melanin fluorescence is an informative predictor for molecular processes in the melanin microenvironment in the melanosomes and their changes in the process of malignant transformation into melanoma cells. Finally, suggestions for further investigations on current issues of melanin spectroscopy using nonlinear spectroscopy are given.

## 2. Method of Dermatofluoroscopy/Objects of Investigations

In conventional spectroscopy, melanin is transferred from the ground state to the fluorescent excited state by absorption of one (UV) photon (Figure 1a). However, besides melanin in the skin tissue, this UV excitation also causes other organic molecules such as NADH/NAD(P)H, β-carotene, hemoglobin and flavins to fluoresce. The fluorescence quantum yield of the latter surpasses that of melanin by more than 3-fold. Therefore, background autofluorescence is dominant and the melanin fluorescence is masked and thus becomes undetectable [5].

A strategy to escape from this dilemma is shown in Figure 1b: The melanin fluorescence is generated by means of a *stepwise* excitation by two photons of lower energy (800 nm). Melanin is first brought into an intermediate state and from there into the fluorescent excited state (Figure 1b). Melanin is characterized by a continuum of excited states; during the lifetime of such an excited intermediate state the fluorescent level can be reached by absorption of a second photon. This results in the melanin fluorescence. It is worth emphasizing that the excitation wavelength of 800 nm excites both eumelanin and pheomelanin [20]. On the other hand, molecules that have no further excited state between the ground state and the fluorescent state—like the above-mentioned endogenous fluorophores—cannot be excited to fluorescence by these low-energy photons. That is the purpose of this special type of excitation: there is virtually no autofluorescence. The latter applies “cum grano salis”: by the process of *simultaneous* two-photon excitation via a virtual intermediate state (Figure 1c), also the endogenous fluorophores can be excited by the 800 nm photons to their fluorescent state. But when using nanosecond pulses, which have low photon flux densities at intensities that are tolerable for tissue, this process has a very low probability; therefore the autofluorescence is weak and does not mask the melanin fluorescence.

This stepwise two-photon excitation as schematically shown in Figure 1b is the principle of dermatofluoroscopy [13,14,15]. In the experimental setting of dermatofluoroscopy, laser pulses with a wavelength of 800 nm and pulse duration of 2 ns are used. Each measuring pulse analyzes a tissue area with a diameter of 30 µm. The lesion being examined is scanned along a measurement grid; a meaningful step size lies between 50 μm and 200 μm. Even with a step size of 200 µm, several hundreds of fluorescence spectra result per lesion. The novel technology of dermatofluoroscopy as a diagnostic method to detect melanoma [10,11,12] has been studied in a multicenter clinical study (study acronym FLIMMA) at the University Dermatology Clinics of Tübingen, Heidelberg and Charité Berlin [16,17,18,19] As a result of this pivotal study the device received its CE-mark and was approved for the medical market. Details pertaining to the device are described in [10,11,17]. Previous fluorescence measurements were carried out in the following settings:

(i) In vivo on pigmented lesions of the skin, (ii) ex vivo on surgically removed specimen of suspicious lesions, and (iii) on the corresponding histological FFPE preparations (formalin fixed, paraffin embedded). The entire database of measurements consists of several tens of thousands of spectra.

The examined individuals were of the Caucasian skin type (Fitzpatrick skin types 1–4). Most individuals of the database were patients of a dermatological practice in Berlin, or the dermatological clinic of the St. Josefs Hospital of the Ruhr University, Bochum. Freshly excised specimen and FFPE preparations originated from patients of the Dermatology Clinic at Vivantes Klinikum Berlin-Südost. In all cases, written informed consent of patients was obtained.

## 3. Fluorescence of Normal Skin, of Benign or Atypical/Dysplastic Nevi and of Melanoma

### 3.1. In Vivo

The most important result of the investigations into several hundreds of samples of pigmented skin tissue, i.e., tens of thousands of measuring areas, is that there are only four types of fluorescence spectra. These spectra belong to the four different pigment cell types: melanocytes, nevomelanocytes, dysplastic pigment cells, and melanoma cells [10,13,17,20]. There are clear differences *between* these four types of spectra, but *within* each mentioned cell type the spectra are largely identical.

In Figure 2, these four characteristic types of spectra are shown (valid for Fitzpatrick skin types 1–4, Caucasians).

The fluorescence spectrum of melanin in melanoma cells is given in Figure 2, lower right (class 1): It shows a uniformly increasing intensity over the entire measurement range of 430–650 nm. This spectral shape results (i) for all melanoma cells when measured on a melanoma, and (ii) for all melanomas of the examined subtypes so far: SSM, ALM, NM, LMM, desmoplastic and hypomelanotic melanoma [20,24,25], and (iii) both for melanoma developing on a preexisting nevus and for de novo melanoma.

This uniform spectral fluorescence shape of melanoma-derived melanin is mathematically characterized in that the first derivative of the intensity course I(lambda) over the wavelength lambda, dI(lambda)/d(lambda), the gradient, is a straight line over the entire range of 430–650 nm.

The uniform fluorescence spectrum of melanin in atypical pigment cells (e.g., in dysplastic nevi) is shown in Figure 2, lower left (class 2). Similar to melanin derived from melanoma cells, there is a monotonously increasing spectral curve from short to long wavelengths, which flattens out towards the red end, at around 570 nm. Mathematically, this curve is characterized by the fact that there is a decrease in the constant value of the first derivative dI(lambda)/d(lambda) above 570 nm. Figure 2, upper right (class 3), shows the uniform fluorescence spectrum of melanin in the nevomelanocytes of benign nevi: a flat band with a maximum between 530 nm and 550 nm. This means, in the course of dI(lambda)/d(lambda), there is a zero crossing between 530 nm and 550 nm. Figure 2, upper left (class 4), shows the uniform melanin fluorescence spectrum of melanocytes: a more distinct band peak at about 490 nm, i.e., a zero crossing of the first derivative at about that wavelength.

With an automated evaluation of measured fluorescence spectra, with a fixed root mean square error (RMSE), an automatic assignment to one of the four types of spectra is possible [13,17,19,25].

The ex vivo fluorescence spectra of melanin in the four pigment cell types (ex vivo surgical specimen)* correspond to the in vivo spectra (see above, Figure 2). In particular, the spectral type for melanoma cells shown in Figure 2, lower right also occurs in surgically removed specimen of all melanoma subtypes listed above.

### 3.2. Fluorescence Spectra of Melanin in the Four Pigment Cell Types in Histological FFPE Preparations (Thin Sections and Histoblocks) *

With the exception of LMM, the spectra in the histological preparations correspond to the four in vivo spectra types. In particular, the type of spectrum shown in Figure 2, lower right, for melanoma cells also occurs in the FFPE preparations of the melanoma subtypes SSM, NM, ALM, and desmoplastic as well as hypomelanotic melanoma (see examples in Figure 3) [24].

In connection with the fluorescence measurement, especially in histological specimen, for LMM and the other melanoma subtypes with a dermal component there is another diagnostic key: the absence of the so-called second-harmonic generation (SHG) of the fluorescence excitation radiation. This means: Whereas in the intact dermis the second harmonic of the excitation radiation (400 nm) is generated in the microcrystalline structures of collagen, the tumorous destruction of the dermis leads to disruption of the microcrystalline structure and thus to failure of the SHG [21,25].

It is worth mentioning that the uniform melanin fluorescence spectrum of melanoma (Figure 2, lower right) also appears in the histological specimen of the choroidal melanoma (Figure 3d) ([26], see also [24]). It should also be noted that the melanin fluorescence spectra of nevi and melanoma of individuals with darker skin types (skin types 5, 6) are clearly different from the spectra derived from individuals with lighter pigmented skin types as shown above [24,25].

* Technical note: Fluorescent markers (e.g., surgical skin markers, dyes used for immunohistological staining) can give rise to fluorescence interfering with the measurements and should be avoided. Nevertheless, spectra of such artifacts are recognized in the automatic evaluation of the fluorescence data and the corresponding tissue sample is removed from a diagnostic evaluation.

## 4. Diagnostic Utility of the Melanoma Fluorescence (Summary)

### 4.1. In Vivo

Early detection of all melanoma subtypes, and for both, de novo melanomas as well as those developed on a pre-existing nevus.

Perspective of non-invasive follow-up of suspicious pigmented lesions.

Characterization of the malignant potential of eruptive nevi (e.g., pregnancy-related or treatment-related cases).

Presurgical definition of incision line for complete excision of melanoma lesions.

### 4.2. Ex Vivo

Optimal tailoring of paraffin-embedded specimen for the preparation of histopathological slides, especially in lesions with eccentric positioning of the most suspicious region.

### 4.3. On the Histological Preparation (FFPE)

Objective assistance or second option for the dermatohistopathologist to arrive at a final histopathologic diagnosis (the present gold standard of melanoma diagnostics) for all melanoma subtypes except LM/LMM. Dermatofluoroscopy may particularly support dermatopathologists in the diagnosis of melanocytic lesions with uncertain malignant potential, e.g., MELTUMP or STUMP; atypical congenital nevi of childhood; or in cases of trans-epidermal melanocytic migration (TEMM, pagetoid spread [22]). Based on the pattern of SHG loss across the histological (FFPE) specimen it is even possible to conclude on the level of invasiveness of melanoma. As mentioned above, the signal of SHG is lost when melanoma cells have invaded dermal layers beyond the basement membrane. This is especially useful for differentiating invasive lentigo maligna melanoma from in situ lentigo maligna and could provide an objective measure for the histopathologists [21,25].

### 4.4. Possible Pitfalls Related to the Uniform Melanoma Fluorescence Spectrum

Fluorescence spectra as shown in Figure 2, lower right, for melanoma cells can also occur in inflamed nevi and after the use of cosmetic skin tanning agents. From a clinical point of view, this is not a very relevant limitation. However, the occasional occurrence of this type of spectra in pigmented seborrheic keratosis or pigmented basalioma requires further investigation. In the strictest sense, fluorescence spectra as shown in Figure 2, lower right, are therefore not an exclusive fingerprint of a melanoma, i.e., the relationship between spectra of Figure 2, lower right and melanoma is not unique. However, this relationship is unambiguous in the sense that every melanoma cell in vivo and ex vivo shows this spectrum of Figure 2, lower right. Most importantly, a lesion that does not show this type of spectral fluorescence when examined in vivo or ex vivo is not a melanoma.

## 5. The Melanin Fluorescence as a Measure for the Process of Malignant Melanocytic Transformation

Dermatofluoroscopy reveals pronounced spectral changes in the melanin fluorescence of the four types of pigment cells, particularly when ranking these according to the process of malignant transformation (see Figure 2): from a fluorescence band in the green spectral range with a maximum at 500 nm in normal melanocytes to a monotonously increasing curve in the yellow-red spectral range up to 650 nm (limit of the measuring range) for melanoma cells. Remarkably, according to the current state of knowledge, the corresponding melanin *absorption* spectra of the four cell types are largely identical: they show a monotonously decreasing curve over the entire visible spectral range towards the red.** Thus, the fluorescence in the process of malignant melanocytic transformation shows an increasing red shift compared to the absorption (so-called Stokes shift). What conclusions can be drawn from this observation about molecular processes?

The melanin fluorescence of cutaneous pigment cells has its origin in melanin oligomers. These oligomers are the fluorescent units. They consist of 3–6 sub-units [27], which are composed of the monomer building blocks of eumelanin or pheomelanin and their respective quinones [27,28]. About 10^6^ of such melanin oligomers form what is known as granules. These melanin granules are attached to a fibrillar matrix, localized in melanosomes bound to the lipid membrane. The melanin oligomers provide the pi-electron system, which determines the spectroscopic properties considered here.

### Internal and External Causes for a Fluorescence Red Shift with Respect to Absorption

In ordinary organic molecules with pi-electron systems, the cause of a redshift of fluorescence compared to absorption is a vibrational relaxation in the final state of the electron transition, the so-called Franck–Condon (FC) state. This relaxation is triggered by the fact that in the FC state the core configuration corresponds still to that in the ground state, but the electron density distribution is already changed. This vibrational relaxation leads to an energetic lowering of this excited state. The fluorescence resulting from this relaxation is red-shifted compared to the absorption.

For molecules in solution, additional energetic lowering of the fluorescing excited state is possible due to a changed interaction with the solvation shell. This so-called solvatochromism can lead to large Stokes shifts of up to several hundreds of nanometers. This phenomenon can also be observed with dissolved melanin, and with a different shift with different solvents, e.g., fluorescence maximum in H_2_O at 510 nm, in DMSO at 575 nm [5].

In the case of coupled oligomers as the origin of the melanin fluorescence in pigment cells, instead of the solvation shell there is the complex microenvironment (protein matrix) of the melanosomes [21]. A red shift of the fluorescence can be caused by FC-relaxation in the pi-electronic excited states of the oligomers as well as by a “reorganization” of the microenvironment. Since the absorption profiles of the four cell types are identical in the relevant range around 800 nm [24,25,28,29,30,31,32], the extent of FC relaxation is identical in all four cell types. In conclusion, the increasing redshift of the melanin fluorescence in the process of malignant transformation from normal melanocytes to melanoma cells is caused by changes in the local microenvironment of the melanin in the melanosomes. In the case of melanoma in particular, this specific microenvironment exists in all subtypes regardless of their genetic and anatomical characteristics and their eumelanin/pheomelanin ratio. There are (at least) two possible causes for such changes:

(i) a minimization/elimination of the interaction of the melanosomes with the keratinocytes via the dendrites in the sequence from melanocytes via normal and atypical nevomelanocytes to melanoma cells [33] and/or.

(ii) a gradual dissolution of the melanosome membrane in in the process of melanomagenesis [27,34].

At present there are not enough experimental results to specify the various mechanisms, but the following findings can be helpful: (i) The robustness of the fluorescence spectra with respect to the dehydration in the process of the histological preparation of tissue specimen (see above), (ii) the even further red shift of the fluorescence in skin types 5 and 6 [25].

** This melanin absorption represents a remarkable exception to the usual structured band spectra of organic molecules and pi-electron aggregates. According to current knowledge, this monotonous melanin absorption in the visible and near-infrared spectral range is the envelope of sub-bands [29]. This is supported, for example, by the fact that the fluorescence spectrum is dependent on the excitation wavelength [5,6].

## 6. Conclusions

With the method of dermatofluoroscopy (stepwise two-photon excitation), the ultra-weak melanin fluorescence of pigmented skin tissue can be measured selectively and spectrally resolved in the visible spectral range. It shows that (i) the melanocytes of normally pigmented skin, the nevomelanocytes of benign and atypical (dysplastic) nevi and the melanoma cells each show a distinct spectral pattern of fluorescence, and (ii) the melanin spectrum of the melanoma cells is independent of the melanoma subtype.

This novel method of dermatofluoroscopy is well suited to diagnose melanoma. It can be used both on patient skin (in vivo) and on freshly excised or formalin-fixated and paraffin-embedded surgical specimen (ex vivo). The four characteristic melanin fluorescence spectra are measures of the molecular structures of the fluorescent pi-electronic melanin units and their relaxation processes. These are influenced—in characteristically different ways—in the four types of pigment cells by the microenvironment in the melanosomes. For further basic research on the molecular processes of malignant melanocytic transformation, it is recommended to use the following methods of nonlinear spectroscopy on each of the four different pigment cell types: (i) Investigation of stepwise as well as simultaneous two-photon fluorescence excitation spectra of melanin with variable excitation wavelength in the red/near infrared region, (ii) hole-burning absorption spectroscopy [35], preferably also in the near-infrared spectral range.

Such investigations could elucidate fluorescence-relevant differences in the subband structures of the melanin absorption spectra. The following examinations using dermatofluoroscopy could contribute to the further qualification of this method for melanoma diagnosis: (i) Inclusion of less frequent melanoma subtypes, e.g., mucosal melanoma [36], (ii) comparative studies on lentigo (L.) simplex and L. solaris versus L. maligna and L. maligna melanoma, with the aim to support the clinical differentiation of these entities that show overlapping morphology.

A transfer of available research results concerning melanin fluorescence is currently under way to further elucidate its role in Parkinson’s disease as well as in human tissue containing neuromelanin [37]. Other promising areas of application for the method of selectively measuring melanin fluorescence are currently under development in various branches of natural science and technology [25], because melanin is found in many animals and also plays a role in bacteria and fungi [38]. Finally, melanin-based nanoparticles have recently become an important field of research in biotechnology [39].

## Figures and Tables

**Figure 1 ijms-22-05265-f001:**
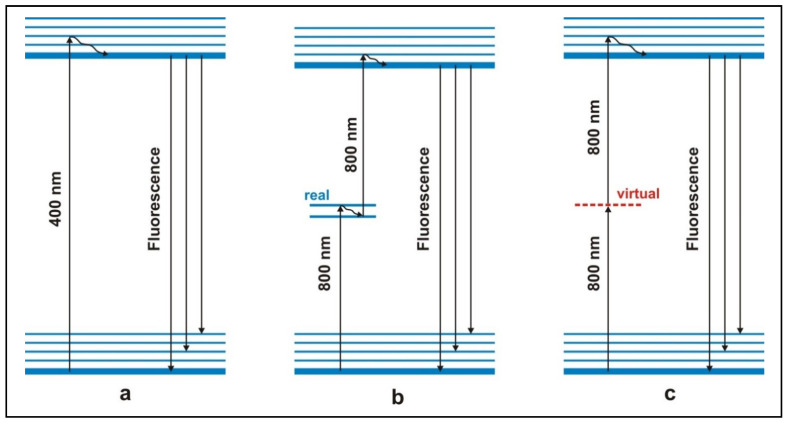
Different modes of excitation of fluorescence. (**a**) Left: conventional fluorescence excitation by one photon (e.g., 400 nm). (**b**) Middle: excitation by stepwise absorption of two photons via a real intermediate energy level (e.g., 800 nm, preferably from a nanosecond laser). PRINCIPLE OF DERMATOFLUOROSCOPY. (**c**) Right: excitation by simultaneous absorption of two photons via virtual energy level (e.g., 800 nm, preferably from a femtosecond laser). This mode of excitation is used in femtosecond laser spectroscopy. Due to the small cross-section, a nanosecond (ns) pulse excitation with tolerable intensities gives rise to only an extremely weak fluorescence.

**Figure 2 ijms-22-05265-f002:**
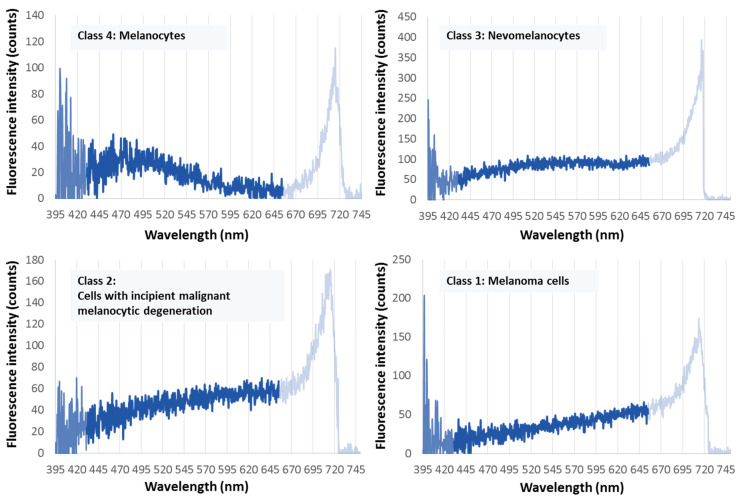
The four characteristic melanin fluorescence spectra of pigmented skin cells in the spectral range between 430 nm and 650 nm. Upper left: melanocytes; upper right: nevomelanocytes; lower left: dysplastic nevomelanocytes; lower right: melanoma cells. The fluorescence is excited by stepwise two-photon absorption with 800 nm/2 ns pulses (principle of dermatofluoroscopy). (The intensity above 650 nm results from another non-linear process not considered here. In any case, it is ensured that its intensity is zero below 650 nm.)

**Figure 3 ijms-22-05265-f003:**
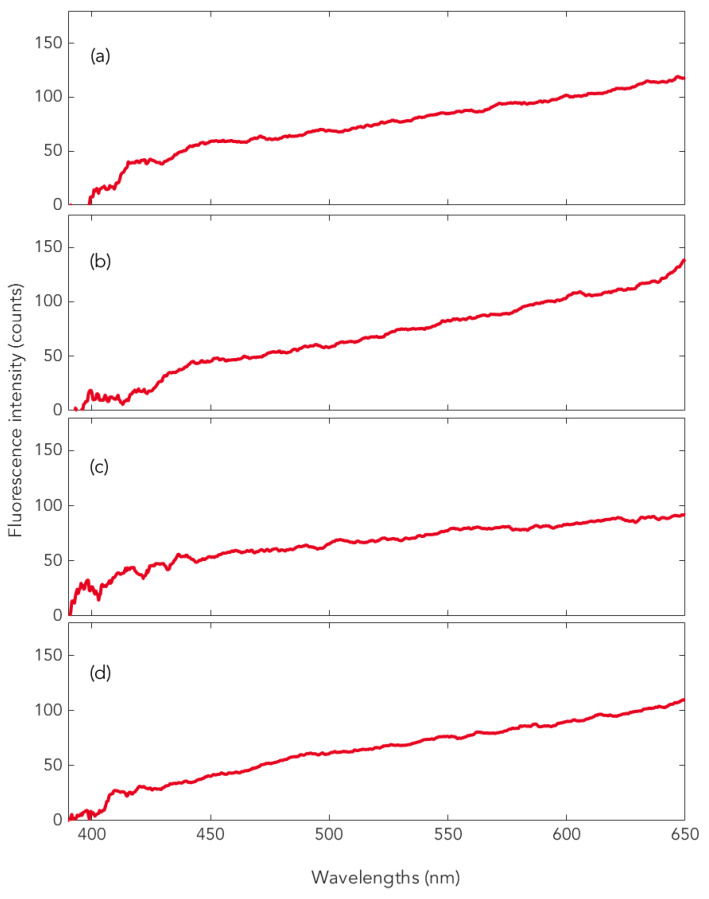
Representative measurements on different types of melanoma: (**a**) SSM, (**b**) NM, (**c**) ALM, and (**d**) choroidal melanoma, all measurements performed in histological specimen (FFPE).

## Data Availability

This review doesn’t present new experimental data. For data availability please refer to the respective original papers.

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
