# Peer review of "From Melanocytes to Melanoma Cells: Characterization of the Malignant Transformation by Four Distinctly Different Melanin Fluorescence Spectra (Review)â€"

_ijms, 2021, doi:10.3390/ijms22105265_

Round 1
Reviewer 1 Report
This interesting report focuses on the fluorescent properties of human melanin pigmentation. This is an important, natural, "biomarker" since melanin is essential for protection against certain forms of UV irradiation, however is also an important indicator or cancerous melanoma. Here, the authors provide a review of melanin fluorescence for a spectrum of melanin types drawn from cells that range from healthy to diseased. Additional information provides conclusions around the possible applications and etiologies of melanin fluorescence. Overall, I found this review helpful and provide a few suggestions for the authors to consider.
1) Several aspects of the manuscript suffer from improper formatting and/or syntax. For instance, L39 begins a single-sentence paragraph. Please correct all syntax and formatting to standard practice.
2) L58: "specialized" or "specialised"
3) L106: the full use of capitalization reads awkwardly here, also this is a single-sentence paragraph.
4) Formatting: (L145) Once again, it is distracting and inappropriate in my opinion to provide dashed lines or bullet points for a scientific article. This section must be rewritten for clarity and standard formatting (i.e., a narrative description in the narrative sections, e.g., "Results", unless the material is moved to a table).
5) Section 3.2 consists of two brief sentences. It is unclear to me why the authors are presenting their information in this manner. Please correct.
Overall, the formatting issues for this manuscript were distracting and non-standard. I encourage the authors to correct their formatting throughout the manuscript.
The scientific material presented is clear and appropriate for publication as a review.
Author Response
The authors would like to thank Reviewer 1 for the careful review and detailed critical hints on unsuitable formatting and syntax. These have all been taken into account and the entire manuscript has been revised accordingly. The entire manuscript has also been revised in the English style as requested. Please see the attachment for the revised manuscript

Reviewer 2 Report
An interesting review about the use of melanin fluorescence as a diagnostic tool for melanoma and pigmented lesions. I have some queries:
Conclusions should be expanded highlighting the possible uses of this technique, and also possible future ideas for studies.
A material and methods section describing how studies were selected (what kind of search engine and what keywords were used) is in my opinion mandatory for this type of review.
Page 2 line46/48 "These subtypes also differ with respect to the prognosis and outcome for the patient, and in terms of potential and localization tendency of metastasis. " this sentence needs a reference, such as https://doi.org/10.3390/medicina57040359
Thank You
Author Response
The authors thank reviewer 2 for critical and helpful suggestions. They were all taken into account in the revision of the manuscript. In detail: The conclusions were expanded in the suggested sense The procedure for selecting the publications is described in a new „Addendum“ section Special thanks for pointing out the new one in ref. [36] cited work
Please see the attachment (revised version)

Round 2
Reviewer 1 Report
The author's response does not detail the changes to the manuscript, beyond a vague suggestion that they have 'revised the manuscript accordingly'. A major revision generally requires more substantive consideration of the critiques raised in original review. I note that several suggestions made in the original review (removing dashes, bullet points, incomplete sentences) have not been addressed.
Author Response
Thank you! The entire manuscript has been revised not only in the English style but now also in formatting as requested. A version with the corrections indicated is attached
Reviewer 2 Report
The authors responded to all queries. The paper is eligible to be published
Author Response
Thank you!